# Synergistic Effect of Yak Dung Fiber and Yak Dung Ash on the Mechanical and Shrinkage Properties of Cement Mortar

**DOI:** 10.3390/ma16020719

**Published:** 2023-01-11

**Authors:** Xuwei Zhu, Lihui Li, Bo Tian, Panpan Zhang, Junjie Wang

**Affiliations:** 1Institute of Highway Science, Ministry of Transport, Beijing 100088, China; 2The Key Laboratory of Road and Traffic Engineering, Ministry of Education, Tongji University, Shanghai 201804, China; 3Yunnan Provincial Institute of Highway Science and Technology, Kunming 650051, China

**Keywords:** yak dung fiber, yak dung ash, cementitious composites, physical and mechanical properties, scanning electron microscopy, energy dispersive X-ray spectroscopy

## Abstract

The high value use of agricultural and livestock waste resources in the context of a low carbon economy is a challenge that currently plagues many countries. Yak dung, as a waste resource from livestock farming in the plateau, is considered to be a misplaced treasure. In this work, yak dung was processed into yak dung fiber (YDF) and yak dung ash (YDA), respectively, and the microscopic morphology of the YDF and YDA was assessed using scanning electron microscopy (SEM). The elements in the YDA were analyzed by energy dispersive X-ray spectroscopy (EDX). Moreover, cementitious composites were prepared with YDF at 0%, 0.3%, 0.5% and 0.7% and by replacing cement with YDA at 5%, 10% and 15% to assess the workability, mechanical properties and shrinkage properties of cementitious composites containing different YDF types (alkali treated and untreated), contents and different YDA contents. The results showed that alkali-treated YDF and YDA contain a large number of honeycomb structure pores, and the strength of cementitious materials with alkali-treated YDF was higher. The addition of YDF had a negative effect on the fluidity and compressive strength of the cementitious material, but the 0.3% YDF was beneficial in increasing its flexural strength. The compressive strength and flexural strength first increased and then decreased with the increase of YDA content. Both YDF and YDA inhibited the shrinkage of the cement paste, and the shrinkage strain of the cement matrix composites containing 0.3% YDF and 10% YDA was reduced by 51.1% compared to the control group. This work is expected to promote the application of yak dung waste in cementitious materials.

## 1. Introduction

A low-carbon economy has become an important trend for sustainable development of today’s society in the grim context of global warming [1]. Controlling and reducing carbon emissions is a major issue that every country needs to face [2,3]. Since infrastructure development is a major contributor to carbon emissions, how to manage the relationship between rapid infrastructure development and the requirement to control and reduce carbon emissions is a crucial issue facing countries [4]. Cement, as an important raw material in the construction and maintenance of infrastructure, emits approximately 0.9 tons of CO_2_ into the atmosphere for every ton of cement produced, and as the second largest carbon emitting industry after steel, it is critical to achieving global CO_2_ emission reduction targets [5,6]. Taking China as an example, 10.4 billion t of semi-rigid base materials are needed for annual road construction, including about 500 million tons of cement, generating about 450 million tons of carbon emissions. According to statistics, the CO_2_ emissions from cement production in 2020 will be about 2.4 billion tons in China [7]. Therefore, reducing the use of cement and improving the durability of cementitious materials are effective ways to reduce carbon emissions on the premise of ensuring concrete strength [8,9]. A common method in reducing the use of cement is to use industrial and agricultural waste resources to replace parts of the cement as a supplementary cementitious material, which not only achieves waste reuse, but also reduces carbon emissions, and is one of the current research hotspots in the concrete industry. The commonly used alternative cementitious materials are sawdust ash, cow dung ash, rice husk ash, fly ash, granulated blast furnace slag, etc. [10,11]. On the other hand, early drying shrinkage or volume changes in cement concrete can easily lead to micro-cracks and affect its durability [12]. Generally, the method of improving the early crack resistance of cementitious materials is to incorporate fibers into the cementitious material. Commonly used fibers include steel fibers, synthetic fibers and plant fibers, of which plant fibers have attracted a great deal of attention from researchers due to their low price and wide availability [13,14].

With the increased government investment in infrastructure construction on the Tibetan plateau in China, a large amount of cement is bound to be consumed during the construction process. Furthermore, due to the special climatic environment of the Tibetan plateau, higher requirements are placed on the durability of the cementitious materials [15]. The yak is a unique herbivorous cattle species living on the Tibetan plateau, and there are approximately 14 million yaks in China [16]. A mature yak typically excretes about 1.68~2.22 kg of yak feces every day, producing approximately 23.5~31.1 million kg of yak feces per day [17]. In the Qinghai Tibet Plateau, yak dung is mainly used by herdsmen for heating and cooking. Overcapacity leads to a large number of yak dung being piled up and put on pasture. The slow decomposition of yak manure leads to excess manure on the surrounding land, adding to the pollution of soil and water systems. Moreover, the raw manure releases harmful gases that can reduce air quality. However, previous studies have shown that the chemical composition of cattle manure is usually composed of 12~13% CaO, 0.8~1.0% MgO, 18~22% Al_2_O_3_, 18~22% Fe_2_O_3_ and 58~62% SiO_2_ [18]. In the meantime, cattle manure contains 14~28% cellulose, 12~22% hemicellulose, 6~13% lignin and 40~55% detergent fiber [19]. It can be seen that cattle dung contains a large amount of cellulose and volcanic ash character material, which is an important reinforcing material in cementitious composites. Therefore, the effective recycling of yak dung waste and its application in infrastructure construction is a solution in line with the concept of sustainable development.

Some efforts and attempts have been made to maximize the value of cattle manure waste in applications such as wastewater treatment, composting, fermentation, porous activated carbon powder, low-cost insulation, road base soil stabilizer, fiber reinforcement and auxiliary cementitious materials [20,21,22,23,24,25,26]. Generally, cattle dung fiber and cattle dung ash are often added to the matrix materials as reinforcement phases in the field of materials and civil engineering, and there is little research on the performance and durability of cattle dung fiber and cattle dung ash alone, but more efforts have been made to study the performance and durability of reinforcing materials. As cattle dung fiber can be obtained by washing and drying cattle dung, its main components include cellulose, hemicellulose and lignin. Chaturvedi et al. studied the mechanical properties of a new polymer composite membrane composed of cattle dung fiber and polyvinyl alcohol (PVA). The results show that alkali treated cattle-dung -fiber-composite-film (CDFC) was having increased bounding and reduced fiber pullout resulting in superior mechanical properties [27]. Ma et al. investigated the tribological and physical-mechanical properties of cattle dung fiber-reinforced friction composites, and the results showed that the cattle dung fiber-reinforced friction composites had more stable friction coefficients and more significant wear resistance [19]. Fasake provided an overview of the potential value of cattle dung fibers in various industrial applications, which have the potential to be a sustainable source of manure fibers as an alternative material for various applications due to the good properties of cattle dung fiber reinforcements [28]. The above literature review proves that cattle dung fibers can not only remove hemicellulose and lignin and increase cellulose content after pretreatment, but also solve the problems of low durability and poor adhesion, and then improve the mechanical properties and durability of their reinforcing materials. Li et al. investigated the strength and self-shrinkage properties of cow dung fiber-reinforced cementitious composites and found that the increase in cellulose content of CDF after alkali treatment led to its tensile strength enhancement, and the 3 d compressive and splitting tensile strengths of cementitious materials incorporated with cow dung fibers, pretreated with 1 mol/L NaOH solution, increased by 21.8% and 12.5%, respectively, and that both cow dung fibers with and without pretreatment inhibited the self-shrinkage in cementitious materials [26].

Cattle dung ash, obtained by calcining sun-dried cattle dung, has volcanic ash properties and is considered to be an effective alternative to cement as a supplementary cementitious material. As far as the application of cattle dung ash in cementitious materials is concerned, lots of effort has been made by researchers. Sahin et al. studied the chemical properties and pozzolanic effect of cattle waste ash, and the content of SiO_2_+Al_2_O_3_+Fe_2_O_3_ in cattle dung ash reached 52.26% and the compressive strength of concrete decreased gradually with the increase of cattle dung ash admixture, but the 56-d compressive strength of concrete with 5%, 10% and 15% cattle dung ash reached 96%, 95% and 94% of the control group, respectively [29]. Ojedokun used cattle dung ash to replace 10%, 20% and 30% cement to prepare concrete. It was found that with the increase of cattle dung ash percentage, the initial-setting and final-setting time increased, and the compressive strength decreased [30]. Mbereyaho et al. studied the effect of cattle dung ash on the strength of mortar and concrete and showed that the compressive strength of mortar specimens with the addition of cattle dung ash first increased and then decreased and that the optimum content of cattle dung ash was 20% [31]. Ramachandran studied the mechanical properties and durability of concrete modified with cow dung ash exposed to fresh water and found that the pH, mechanical properties, durability and antimicrobial activities of fresh water-exposed concrete modified with CDA were better, and the optimum content of cow dung ash was 15% [32]. The above literature review indicates that cattle dung ash has the potential to be used to develop more durable and sustainable high-performance concrete, but there is disagreement in different studies on the optimal amount of cattle dung ash to be incorporated in concrete, which may be related to the conditions of preparation of cattle dung ash as well as the dietary structure and individual differences of cattle [33].

In summary, there is still very limited research into the use of cattle dung fibers in cementitious materials, and there is no report on the synergism application of cattle dung fiber and cattle dung ash being used as reinforcement materials in cementitious composites. The main objective of this study is to investigate the effect of yak dung fibers (YDF) and yak dung ash (YDA) on the mechanical and shrinkage properties of cement mortar when they act synergistically. For this purpose, the effects of different YDF types (alkali treated and untreated), contents (0%, 0.3%, 0.5% and 0.7%) and different YDA contents (0%, 5%, 10% and 15%) on the flowability, compressive and flexural strengths at 3 d, 7 d and 28 days and shrinkage properties of cementitious composites up to 7 d after demolding were evaluated, and the analysis of variance (ANOVA) was also used to statistically analyze all the test results to determine the significance of the effects of YDF type, YDF content, and YDA content on the mechanical and shrinkage properties. Furthermore, the surface morphology of alkali-treated and untreated YDF, YDA and cementitious materials at the curing age of 28 d were analyzed to determine the mechanism of the synergistic effect of YDF and YDA on the physical and mechanical properties and shrinkage properties of cementitious materials. This work is expected to promote the application of yak dung waste in cementitious materials and to improve its crack resistance.

## 2. Experimental Section

### 2.1. Materials

The ordinary Portland cement P·Ⅰ 42.5 (OPC), conforming to the Chinese standards, produced by Qufu Zhonglian Cement Co., Ltd, Shandong, China. was used in this study. The chemical composition of cement determined by X-ray fluorescence analysis (XRF) was shown in Table 1. The physical properties of the cement were tested according to the specification JTG 3420-2020 [34], and the results are given in the Table 2. The fine sand is ISO standard sand, purchased by Xiamen Aisiou Standard Sand Co., Ltd, Fujian, China. The physical properties of sands are shown in Table 3. A superplasticizer (SP) is a polymer-based high performance water reducer, with a specific gravity of 1.06 and a color of clear liquid. It satisfies the GB50119-2016 Chinese specifications [35].

The yak dung is obtained from the excrement of yaks, which is taken from the grasslands of the Tibetan plateau pastoral areas. When it is obtained, it has been dried in the sunlight in the form of cakes. Here, since yak mainly feed on grass and highland barley straw, their dung contains a lot of straw fibers. Therefore, yak dung is used to make YDF and YDA.

### 2.2. Treatment of Yak Dung

#### 2.2.1. Preparation of YDF

The cellulose content determines the mechanical properties of the fiber, and a high cellulose content is beneficial to the tensile strength of the fiber. Li et al. treated cattle dung fibers using 1 mol/L and 3 mol/L NaOH and found that the tensile strength of alkali-treated cattle dung fiber was 89 MPa and 84 MPa, respectively, which increased by 36.9% and 29.2% compared to 65 MPa in the control group [26]. This indicates that alkali treatment of cattle dung fiber can improve its tensile strength, which is due to the increased cellulose content and the increase of crystallinity index caused by the removal of hemicellulose, lignin and extracts. However, under the action of high concentration of NaOH solution, the integrity of YDF will be damaged, and its mechanical properties will also be reduced. Therefore, the NaOH solution used to treat YDF was 1 mol/L in this study. Natural air-dried yak dung obtained from Tibet rangeland was soaked in a NaOH solution with molar concentration of 1 mol/L and a volume of 5 L. Stir thoroughly so that the yak dung is completely submerged and dispersed in the NaOH solution, and then soak for 30 min. After soaking, wash with distilled water until the cleaning solution is neutral pH, then drain the aqueous solution to eliminate fatty debris and hemi cellulose. The semi-finished fiber was put into an oven at 60° for 24 h to obtain the treated YDF. Similarly, the natural air-dried yak dung is directly washed with distilled water and then dried to obtain untreated YDF. The water absorption test revealed that the water absorption of treated and untreated YDF was 1% and 1.2%, respectively.

#### 2.2.2. Preparation of YDA

Nitrogen, potassium and calcium are abundant in YDA’s chemical makeup, and the carbon to nitrogen ratio is comparatively high [30]. The yak dung used for this project work is obtained from Tibet rangeland, dried in sunlight. As the main food intake of yaks in the Tibetan plateau area is grass, barley rice husk and barley straw, yak dung contains a large amount of undigested fiber [17]. In order to prevent the crystalline transformation of SiO_2_ in YDA, which leads to a decrease in the content of amorphous silicon and reduces its activity. In this study, referring to the optimal calcination condition of highland barley straw ash, the calcination temperature was 600 °C and the calcination time was 2 h [36]. Therefore, yak dung was calcined in a central temperature-controlled muffle furnace at 600 °C for 2 h and sieved through 425 μm after the temperature had dropped to room temperature to obtain the experimental YDA.

### 2.3. Mix Proportions and Specimen Preparation

In order to investigate the effect of YDF and YDA on the mechanical properties and shrinkage characteristics of cementitious composites, thirteen cementitious mixes were designed in this paper, and the mix composition is shown in Table 4. The YDF includes both untreated and treated YDF at 0.3%, 0.5% and 0.7% by weight of cementitious material, respectively. The YDA was used as a partial replacement for cement in different proportions of 5%, 10% and 15%. The content of binder was the same for all blended systems, while the water to binder ratio (w/b) of 0.4, and the binder to sand ratio (b/s) of 0.5 have been kept constants for all mixes. In addition, the content of SP and the amount of water used in all mixes were also kept the same. As an example, C11-F0.3-A10 indicates a mix numbered C11, with a 0.3% admixture of YDF and a 10% mass percentage of YDA replacing cement, and a C0 control group, with no added YDF or YDA. 

The procedures for the preparation of the YDF cement mortar specimens consisted of the following steps. Firstly, a certain mass of dried YDF was weighed and water was absorbed until saturated. Secondly, the saturated YDF were added to the mixing water (70%) and placed in ultrasound for 3 min to ensure a homogeneous dispersion of the YDF. Thirdly, the mixed fiber solution was added to the cement mortar mixer and slowly added to the cement and mixed for 1 min. Fourthly, the remaining mixing water (30%) and SP were added and mixed for 1 min. Fifthly, ISO standard sand was slowly added and mixed for 2 min. Finally, the freshly mixed cement mortar was filled into the specimen moulds and vibrated concomitantly. And then, each specimen was demolded after being hardened in the laboratory for 24 h. Demolded specimens were placed to a concrete curing room at a temperature of 20 ± 2 °C and a humidity of 97% until the test age.

### 2.4. Characterization and Test

#### 2.4.1. Physical and Mechanical Properties

The effect of the addition of YDF and YDA on the physical properties of cement mortars can be determined by monitoring the fluidity of the fresh cement mortar mix. After obtaining a fresh cement mortars mixture, fluidity test was performed to evaluate the workability of the cement mortar specimens according to the Chinese standard GB/T 2419-2005 [37].

According to the Chinese standard JTG 3420-2020 [34], 40 × 40 × 160 mm prismatic cementitious sand specimens were used to determine their flexural and compressive strengths at standard curing for 3, 7 and 28 days. The flexural strength test was first carried out, after the test, the broken mortar specimens were taken for the compressive strength test. The average of the strength of three parallel mortar specimens is taken as the final strength test result.

#### 2.4.2. Shrinkage Test

The curing age of 1d is used as the starting point for measuring mortar shrinkage in this paper. As shown in Figure 1, demolded 40×40×160 mm cement mortar specimens were used for shrinkage experiments. The shrinkage value was measured using a displacement sensor manufactured by Beijing Huajian Technology Co Ltd., Beijing, China, which had a measuring range of 20 mm and an accuracy of 0.01 mm. The test method is as follows: firstly, one end of the specimen is fixed on a flat surface and the other end is glued to a 50×50×2 mm glass sheet using AB glue. Then, an all-direction bearing is fitted, and the displacement transducer is held in place using the all-direction bearing, and the probe of the displacement sensor is pressed against the glass plate where the displacement transducer is finally connected to a collector controlled by a computer. When all preparations are complete, the sensor displacement value is cleared to zero and the shrinkage change of the cement mortar specimen is monitored in real time. Measurements were carried out in laboratory conditions with a 10-min acquisition interval, and the laboratory temperature was periodically varied with atmospheric temperature. The shrinkage strain of a cement mortar specimen is calculated via the following equation.
(1)$$\varepsilon_{t}=\frac{\Delta l}{l_{0}}=\frac{\Delta l}{160}\times 10^{6}$$
where ε*_t_* is the shrinkage strain of the cement mortar specimen at t moment (μm/m), Δ*l* is the deformation of the specimen at t moment (mm) and *l*_0_ is the length of the cement mortar specimen, 160 mm.

#### 2.4.3. Microstructure Analysis

Microscopic morphology of YDF, YDA and different cement mortar specimens were characterized by scanning electron microscope (SEM) with an accelerating voltage of 5–15 kv, and the specimens needed to be gold sprayed to conduct electricity prior to testing. The elements in the YDA were analyzed by energy dispersive X-ray spectroscopy (EDX).

## 3. Results and Discussion

### 3.1. Surface Morphology Characterization of YDF

The dispersion of 3 g treated YDF and 3 g untreated YDF in water is given in Figure 2. It can be clearly seen that the treated YDF with NaOH solution are more evenly dispersed in the water, indicating that the treated YDF is less prone to agglomerate during the mixing process and that the fibers are more uniformly distributed in the hardened concrete specimen. The SEM images of the YDF surface before and after treatment with NaOH solution are shown in Figure 3. The surface of the untreated YDF was relatively uniform and wrinkled, but the wrinkles are smoother (Figure 3a), and the surface of the treated YDF with NaOH solution appeared hollow, and many grooves can be observed, and their surface texture was rougher (Figure 3b). The reason is that the strong alkaline solution detaches hydrogen attachment between the cellulose molecules in the YDF, and impurities such as hemicellulose and lignin in the YDF are dissolved, which enhances the surface roughness of the fiber [26]. According to Fiore’s findings, a rough fiber surface can improve interfacial adhesion between the fiber and the cement matrix [38]. Therefore, the YDF treated with NaOH solution is more suitable to be incorporated into cementitious materials, whereas the untreated YDF have weak bonding in the interface transition zone between the fibers and the cementitious materials, which may affect the mechanical properties of the cementitious materials.

### 3.2. Characterization of the Surface Morphology and Elemental Composition of YDA

Figure 4 shows an SEM image of YDA. As seen in Figure 4, cow dung maintains the original plant skeleton of YDF after calcination at high temperatures. The surface is plate-like, consisting of many interconnected thin-walled tubules. Its overall structure is loose, honeycomb-like and has a large number of pores, which are on a scale of around 5–10 μm. EDX analysis was performed on the entire area in Figure 4b, and the results are shown in Figure 5. YDA is mainly composed of C, O, Si, Ca, Al and Fe elements. Among them, the content of C element is the most; this is because yak dung is not burned sufficiently in the muffle furnace, which is mainly found in carbon black. The small size of carbon black in relation to cement particles allows it to be dispersed within the cement matrix during the preparation of cement mortars, where it can fill some of the pores and improve the microstructure of the matrix [39]. On the other hand, the elements Si, Al and Ca are mainly present in the form of oxides in YDA and can be directly involved in the hydration reactions of cement. Because the calcination temperature is 600 °C, the SiO_2_ in YDA is amorphous and has a high activity, which can react with Ca(OH)_2_ to produce a higher strength low alkalinity hydrated calcium silicate gel [32].

### 3.3. Fluidity

The fluidity of cement mortars with different contents of YDF and YDA are given in Figure 6. The fluidity of C0-F0-A0 cement mortar is used as a reference value for comparison. It is clear that the fluidity values of the cement mortars with the addition of YDF and YDA are both lower than C0-F0-A0. Among them, the fluidity of cement mortar gradually decreases with the increase of YDF content. This is because the addition of YDF increases the interlocking effect between the cement matrix, resulting in a reduction in its fluidity. At the same time, it can be found that the fluidity value of cement mortar containing pretreated YDF is less than that of the untreated YDF at the same content of YDF. As described in Section 3.1, the alkali-treated YDF have a rougher surface, which improves the interfacial adhesion between the fibers and the cement matrix, resulting in a lower fluidity of cement mortar mixed with alkali-treated YDF. In addition, the fluidity value of C1-F0.3-A0 cement mortar is reduced by 9.9% compared to C0-F0-A0 cement mortar, and the fluidity values of C2-F0.5-A0 and C3-F0.7-A0 cement mortars are reduced by 4.5% and 3.3% compared to C1-F0.3-A0 and C2-F0.5-A0 cement mortars, respectively. The reduction in fluidity value decreased gradually with the increase in the amount of YDF incorporated. This is because as the YDF content increases, the YDF are difficult to disperse uniformly in the cement mortar, agglomeration of YDF in the cement mortar is observed and the bonding effect of the YDF to the cement matrix is weakened, which is manifested by a non-linear reduction in the fluidity of the cement mortar. A similar trend was observed in cement mortars incorporating untreated YDF. It can also be seen from Figure 6 that as the proportion of cement replaced by YDA increases, the fluidity of the cement mortar gradually decreases, with the fluidity values of C7-F0-A5, C8-F0-A10 and C9-F0-A15 decreasing by 8.7%, 17.4% and 27.3% compared to C0-F0-A0, respectively. The reason for this is that the YDA is honeycombed and contains a large number of internal voids, which adsorb a large amount of free water during the cement mortar mixing process, resulting in less water flowing between the particles of the cementitious material. As expected, the addition of both YDF and YDA to the cement mortar resulted in the smallest cement mortar fluidity, with fluidity values reduced by 12.8%, 20.9% and 31.4% for C10-F0.3-A5, C11-F0.3-A10 and C12-F0.3-A15 compared to C0-F0-A0, respectively.

### 3.4. Compressive Strength

The effects of pretreated and untreated YDF on the 3 d, 7 d and 28-d compressive strength of cement mortars are shown in Figure 7. The reduction in the compressive strength of the cement mortar was observed as the content of YDF increased at each age. As an example, the 28-d compressive strength of cement mortars with pretreated YDF reduced by 3.5%, 14.2% and 28.8%, respectively, when the content of YDF was increased from 0.3% to 0.7%. The reduction of strength can be attributed to the uneven distribution of YDF in the cement matrix, resulting in a large number of pores or voids in the cement matrix and a weaker transition zone at the interface between the YDF and the cement mortar, which significantly reduces the compressive strength [40].

It can also be seen from Figure 7 that the addition of pretreated and untreated YDF has a different effect on the compressive strength of the cement mortar. Compared to the control group, the 28-d compressive strengths of cement mortars with pretreated and untreated YDF were decreased by 3.5% and 15.4%, respectively. Meanwhile, the 28-d compressive strengths of cement mortars with pretreated and untreated YDF were reduced by 28.8% and 34.1%, respectively, when the YDF content was increased to 0.7%. It is apparent that the compressive strength of cement mortars with pretreated YDF is greater than that of cement mortars with untreated YDF for the same YDF content at each age. In addition, when the content of pretreated YDF is low, the compressive strength is slightly less than that of the control, and the strength of the cement mortar decreases rapidly with the increase of pretreated YDF content. It is suggested that alkali pretreatment of YDF contributes to strength development since pretreatment reduces the amount of lignin and hemicellulose in YDF and creates a rough surface, thereby enhancing the interlocking effect between the YDF and the cement matrix [26,38]. Furthermore, the compressive strength of cement mortar mixed with YDF grows slowly in the early stages of curing. For example, when the content of pretreated YDF was 0.7%, the 7-d compressive strength of cement mortar increased by 8.6%, while the control group increased by 18.6%. The reason may be related to the composition of YDF in the sense that yak dung can only partially remove lignin and hemicellulose after soaking in NaOH solution, and hemicellulose and lignin will affect the hydration rate of cement [41]. And when the addition of YDF is large, the YDF will dissolve more carbohydrates after contacting with water, thus inhibiting the development of cement hydration and strength [26]. Hence, when YDF are added to cement mortars, it is necessary to treat it with alkali and the optimum addition is no more than 0.3%.

Figure 8 shows the compressive strength of cement mortar specimens with different YDA contents. The compressive strength first increased and then decreased with the increase of YDA content. When the content of YDA was 10%, the compressive strength of 3 d, 7 d and 28 d reached the maximum value, which were 1.22, 1.21 and 1.09 times of the compressive strength of cement mortar without YDA at each age, respectively. When the YDA content is 15%, the compressive strength is reduced but still greater than that of the cement mortar specimen without YDA. This can be attributed to the water absorption effect, active effect and the filling effect of the YDA. During the mixing of the cement mortar, the water–cement ratio of the cement mortar decreases due to the water absorption effect of the YDA, so the compressive strength of the cement mortar containing YDA is greater during the initial stage of curing [42]. With the increase of curing age, the active effect of the YDA begins to take effect and the active components in the YDA, SiO_2_ and Al_2_O_3_, and the Ca(OH)_2_ generated by the hydration of the cement further undergo a secondary hydration reaction to form the calcium silicate (aluminate) hydrate C-S(A)-H gel, which can further reduce the voids and lead to the increased strength performance of cement mortar [32]. In addition, the carbon black in YDA can be dispersed inside the cement matrix with the preparation process of the mortar, which leads to the repair of voids and the improvement of the microstructure of the matrix. However, when the content of YDA is too high, unreacted or residual YDA is mixed in the cement matrix, which will destroy the integrity of the cement mortar and results in the reduced compressive strength. Therefore, the optimum YDA content in cement mortar is equal to 10%.

Figure 9 presents the compressive strength of cement mortars cured at 3 d, 7 d and 28 d with YDF and YDA compounded, where YDF content is 0.3% and YDA content is 5%, 10% and 20%. When the YDF content was 0.3%, and the YDA content increased from 5% to 15%, the 3-d compressive strength first increased and then decreased, reaching a maximum of 41.3 MPa for C11-F0.3-A10 and a minimum of 34.2 MPa for C12-F0.3-A15, which was lower than C0-F0-A0. The compressive strength of all mortars increases with the curing age, with C0-F0-A0 and C11-F0.3-A10 having very approximate 28-d compressive strength, but C12-F0.3-A15 still having a lower 28-d compressive strength than C0-F0-A0. The reason for this is that YDA has a strong water absorption, and the larger the proportion of cement it replaces, the lower the water–cement ratio of the cement mortar mixture, making it difficult to distribute YDF evenly in the cement mortar mixing process, and the agglomeration phenomenon is likely to occur. The end result is that a large number of pores or voids exist in the cement mortar and the compressive strength of the cement mortar is significantly reduced.

### 3.5. Flexural Strength

The relationship between the YDF content (0.3%, 0.5% and 0.7%), the YDF type (pretreated and untreated) and the flexural strength of the cement mortar at different curing ages is given in Figure 10. It can be seen that the type and content of YDF has a significant effect on the flexural strength of the cement mortar. The flexural strength increases with increasing the content of pretreated YDF from 0 to 0.3%. However, the flexural strength decreased when the content of pretreated YDF was increased from 0.3% to 0.7%. In contrast, the flexural strength of the cement mortar gradually decreases as the untreated YDF content increases. This is because the alkali treatment of YDF removes lignin and hemicellulose, which are not beneficial for strength growth, and increases its cellulose content. The high cellulose content favors the tensile strength of YDF, and the bridging effect between YDF and the cement matrix limits the development of cracks, which results in the increased flexural strength of the cement mortar, a finding that is consistent with the findings of Li et al. [26]. On the other hand, the rough surface of the alkali-treated YDF increases the adhesion to the cement matrix. Nevertheless, when the YDF content is excessive, it tends to agglomerate in the cement mortar, resulting in a weak zone of force in the hardened cement mortar. In summary, the flexural strength of cement mortars can be improved when the pretreated YDF content is 0.3% in the cement mortar.

Figure 11 shows the flexural strength of cement mortars with different YDA contents at 3 d, 7 d and 28 d curing ages. A comparison of Figure 8 shows that the flexural strength first increases and then decreases with the increase of YDA content. When the YDA content is 10%, the 28 d flexural strength reaches the maximum value of 10.9 MPa. Therefore, YDA can be used as an auxiliary cementitious material to replace a certain proportion of cement in the concrete mixture.

The flexural strength of cementitious mortars at different ages by the coordinated action of YDF and YDA is shown Figure 12, where the optimum content of YDF was chosen to be 0.3%, and the YDA content was increased from 5% to 15%. As can be seen, the flexural strength of all specimens increases with increasing curing ages. However, the flexural strength first increased and then decreased with the increase of YDA content in each curing age. The 28 d flexural strength reaches a maximum of 10.5 MPa when the YDA content was 10%. Compared to C0-F0-A0, the enhancement of flexural strength at 28 d of 3% and 8.2% can be noted for C10-F0.3-A15 and C11-F0.3-A10, respectively, while C12-F0.3-A15 decreased by 12.4%. It was illustrated that the flexural strength of cement mortar increases to a certain extent when the proportion of cement replaced by YDA is not greater than 10%, while on the contrary, the flexural strength decreases significantly. A possible explanation is that when the proportion of YDA replacing cement is excessive, part of the free water used for free sliding between cement particles is adsorbed by the honeycomb structure of the YDA, as mentioned in Section 3.2, during the mixing of the cement mortar, which minimized the water/cement ratio of the cement matrix and results in the YDF to be difficult to distribute uniformly in the cement matrix and produces agglomeration, which is consistent with what was observed during the tests. The result of YDF agglomeration is the introduction of a large number of harmful pores that are difficult to compact into the cement mortar, which subsequently significantly reduces of the flexural strength.

### 3.6. Shrinkage Test

Reference [43] pointed out that the changes in the volume and length of the specimen can be used to evaluate the shrinkage of cementitious materials, but the results of the current shrinkage test methods are the combination of autogenous shrinkage, dry shrinkage and chemical shrinkage. Autogenous shrinkage is caused by a gradual decrease in the relative humidity within the cement matrix as the cement hydrates, where the capillary state gradually changes from saturated to unsaturated, resulting in an increase in negative pressure in the capillaries. Drying shrinkage is caused by the migration of moisture from within the matrix due to the low humidity of the external environment of the cementitious material. Chemical shrinkage is the change in volume of the hydration early mixture, resulting from the volume of the hydration products being smaller than the volume of the raw material during the hardening process of the cementitious material [44]. Shrinkage tests were carried out on different mortar specimens, and the results are shown in Figure 13. It can be seen that, in general, the shrinkage strain in all mortar specimens increases with increasing test time but increases sharply within 24 h after demolding and slows down after 24 h. Meanwhile, the cyclical changes in laboratory temperature during the day (Figure 14) caused the mortar specimens to show cyclical expansion and shrinkage. The maximum shrinkage occurs at C0-F0-A0, and the minimum shrinkage occurs at C11-F0.3-A10. When shrinking for 156 h, the shrinkage strain of C11-F0.3-A10 is reduced by 51.5% compared to C0-F0-A0, and the shrinkage strain of C12-F0.3-A15 is increased by 27.8% compared to C11-F0.3-A15, indicating that the addition of YDF and YDA is beneficial in improving the shrinkage resistance of the cement matrix. However, when YDF and YDA are used together, the addition of excessive amounts YDA may cause a poor distribution of YDF, which is unfavorable to improving the shrinkage resistance of cementitious materials. There are three reasons for the above test results. Firstly, the addition of the appropriate amount of YDA improves the particle gradation of the powder and increases the compactness of the cement paste. In addition, the participation of YDF in the secondary hydration improves the internal pore structure, reduces the pore content and reduces the outward migration of water, thus providing better volume stability. Secondly, the addition of YDF to the cement paste creates a complex three-dimensional system in which the YDF is randomly distributed and effectively constrains the shrinkage of the cementitious material. Finally, the porous structure of YDF and YDA absorbs a large amount of free water during mixing, which is released when the internal humidity drops, providing an internal curing effect and effectively inhibiting shrinkage.

### 3.7. Statistical Analysis Based on ANOVA

The statistical significance of the experimental results was quantified at a significance level of 0.05. The measured parameters of cement mortar such as fluidity, 28-d compressive strength, 28 d flexural strength and 156 h shrinkage strain were characterized as dependent factors, whereas YDF type, YDF content and YDA content indicated independent factors. The analysis results are presented in Table 5. The *p*-value less than 0.05 classifies a parameter as significant. As can be seen from Table 5, YDF type has a significant impact on 28-d compressive strength, YDF content has significant influence on fluidity, 28-d compressive strength and 156 h shrinkage strain, and YDA has a significant effect on fluidity. In other words, the content of YDF has resulted in a significant decrease in the fluidity, compressive strength and shrinkage strain. At the same time, the content of YDF and YDA has no significant effect on the flexural strength, because there is an optimal amount of cement mortar to maximize the flexural strength.

### 3.8. Scanning Electron Microscope Analysis

Figure 15 presents SEM images of YDF- and YDA-reinforced cement mortars cured for 28 d. It can be seen from Figure 15 that the microstructure of YDF- and YDA-reinforced cement mortar mainly consists of hydrated products, unhydrated particles, YDF, pores and microcracks. Figure 15a illustrates the SEM image of the C0-F0-A0 mortar. As can be seen, more homogeneous C-S-H gels and microcracks at the interfacial transition zone are present in the mortar. Figure 15b shows the SEM image of the cement mortar with 0.3% untreated YDF. It can be seen that the bond between the YDF and the cement matrix is poor, resulting in a defective cement mortar. As shown in Figure 15c, the YDF treated with the NaOH solution was tightly wrapped between the mortar matrix, indicating that the alkali pretreatment of the YDF increased the bond between the YDF and the cement matrix, which is the reason why the strength of the mortar specimens with the addition of the pretreated YDF was greater than that of the untreated specimens. Figure 15d shows the SEM image of the cement mortar with 10% YDA. As can be seen, there are fewer pores and microcracks in the C0-F0-A10 specimen compared to C0-F0-A0. This is because the secondary pozzolanic activity of YDF is further activated, thus more calcium hydroxide crystals are consumed and additional C-S(A)-H gel is generated, which together with the unreacted YDF particles fill the capillaries and microcracks to form a denser microstructure. The SEM images of C0-F0.3-A10 and C0-F0.3-A15 are given in Figure 15e,f. It can be found that the distribution of YDF in the C0-F0.3-A10 specimen is uniform, and the microstructure is dense, and there is a good bond between the YDF and the cement matrix. However, the YDF in the C0-F0.3-A15 specimen is agglomerated and the bond between the YDF and the cement mortar matrix is poor, indicating that when the amount of YDA is excessive, it will adsorb a large amount of free water, which makes it difficult for YDF in cement mortar to be evenly distributed, thus adversely affecting the mechanical properties of the cement mortar.

## 4. Conclusions

In this study, the YDF obtained by pretreatment of cow dung with 1 mol/L of a NaOH solution and the YDA obtained by calcination of cow dung at 600 °C were used in the preparation of cementitious reinforcement materials. The effects of untreated and pretreated YDF and YDA on the mechanical properties and shrinking properties of cementitious composites were studied. The main conclusions are as follows.

(1) YDF treated with the NaOH solution is more evenly dispersed in water, and NaOH can dissolve impurities such as hemicellulose and lignin in YDF, which makes the YDF surface appear hollow and enhances its surface roughness, thus improving the interface adhesion between YDF and cement matrix. 

(2) YDA is composed of honeycomb structures with a scale of about 5–10 μm and contains a large number of pores. YDA contains elements such as Si, Al and Ca in the form of oxides in addition to C. It can react with the hydration product Ca(OH)_2_ of cement in a secondary hydration reaction to produce more C-S(A)-H gels and improve the denseness of cementitious materials.

(3) The fluidity of cement mortar decreases with the increase of YDF and YDA content. The compressive strength of cement mortar decreases gradually with the increase of YDF content, but when the content of YDF exceeds 0.3%, the compressive strength of cement mortar decreased significantly. The compressive strength of cement mortars with pretreated YDF is greater than that of cement mortars with untreated YDF for the same YDF content at each age.

(4) The flexural strength of cement mortar first increased and then decreased with the increase of pretreated YDF content. When the content of pretreated YDF was 0.3%, the flexural strength reached the maximum. However, the flexural strength of cement mortar gradually decreased with the increase of untreated YDF content.

(5) With the increase of YDA content, the compressive strength and flexural strength of cement mortar increased first and then decreased, and when the YDA content was 10%, the compressive strength and flexural strength reached the maximum. When YDF and YDA acted in synergy, the content of YDF was 0.3% and the content of YDA was 10%, and the compressive strength and flexural strength of cement mortar were the largest, which was superior to the physical and mechanical properties of the control group.

(6) The incorporation of YDF and YDA was beneficial to inhibit the shrinkage of cementitious materials. When the content of YDF and YDA in cement mortar is 0.3% and 10%, the shrinkage strain of cement mortar was the smallest, which is reduced by 51.1% compared with the control group at 156 h shrinkage age.

In general, this paper conducted a preliminary study on the recycling of yak feces and demonstrated that both YDF and YDA can be used as an auxiliary additive material in concrete materials, which is of great significance for improving the recycling rate of yak feces and reducing carbon emissions in the cement manufacturing process. Nevertheless, further research is still needed in terms of the optimal preparation process, practical engineering applications, economic benefits and durability of cementitious mixture of YDF and YDA in the future.

## Figures and Tables

**Figure 1 materials-16-00719-f001:**
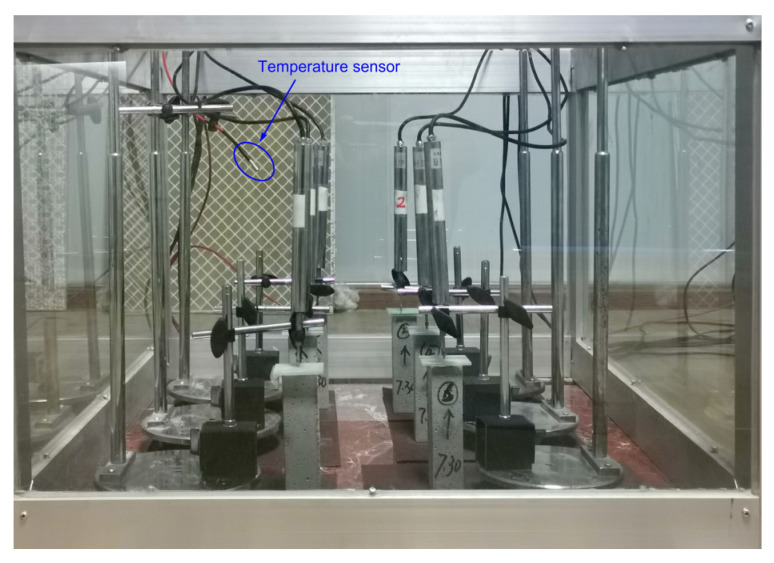
Shrinkage test of the cement mortar specimens.

**Figure 2 materials-16-00719-f002:**
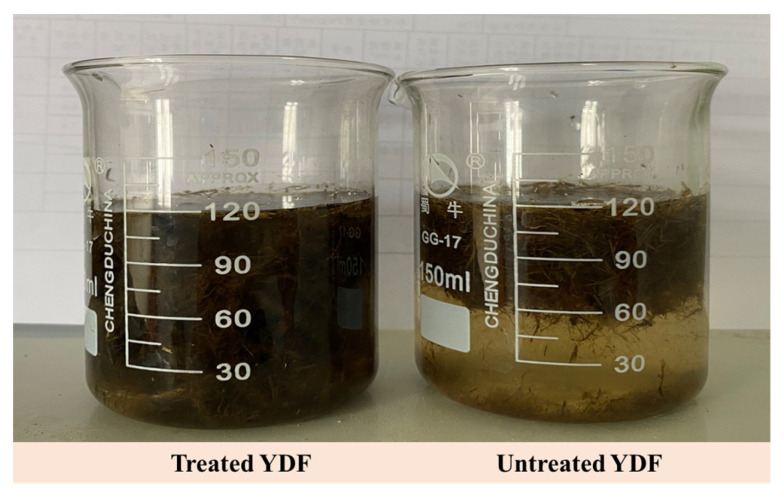
The dispersion of treated YDF and untreated YDF in water.

**Figure 3 materials-16-00719-f003:**
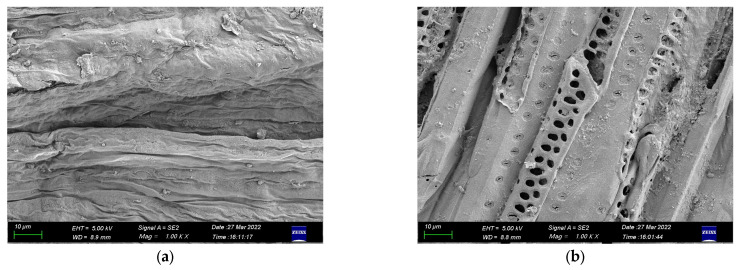
SEM image of the YDF surface before and after pretreatment with NaOH solution: (**a**) untreated YDF; (**b**) treated YDF.

**Figure 4 materials-16-00719-f004:**
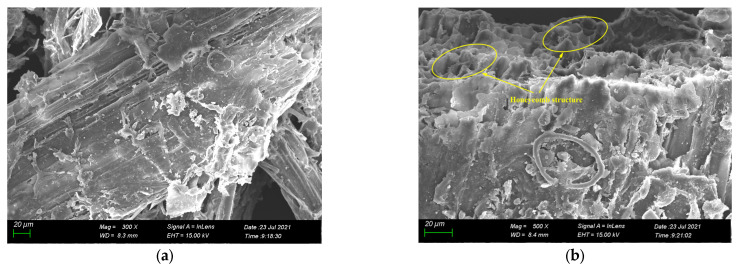
SEM image of YDA at (**a**) ×300 and (**b**) ×500 magnifications.

**Figure 5 materials-16-00719-f005:**
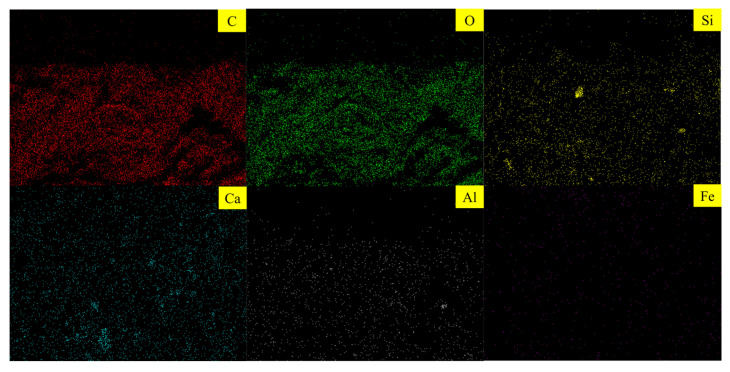
Elemental composition of YDA obtained from EDX.

**Figure 6 materials-16-00719-f006:**
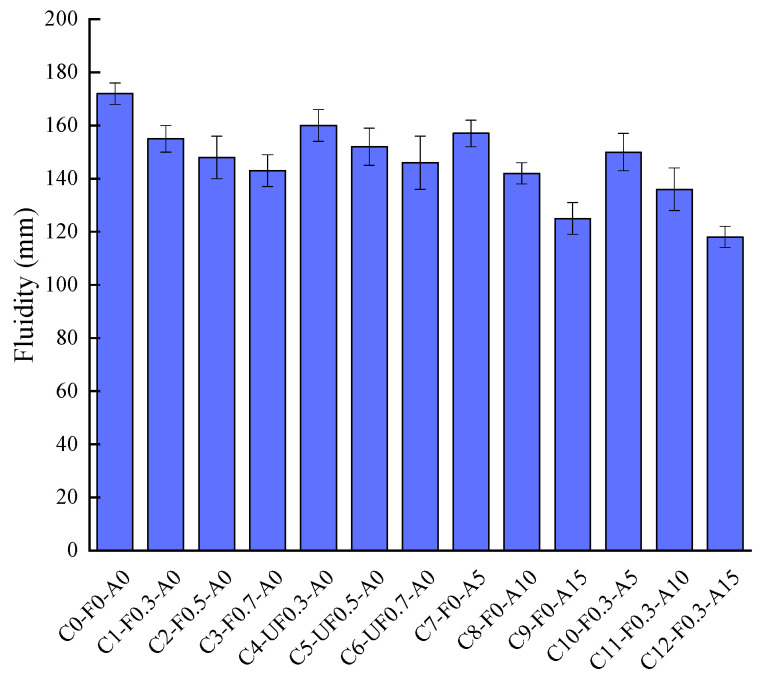
Effects of YDF and YDA content on the fluidity of cementitious mortars.

**Figure 7 materials-16-00719-f007:**
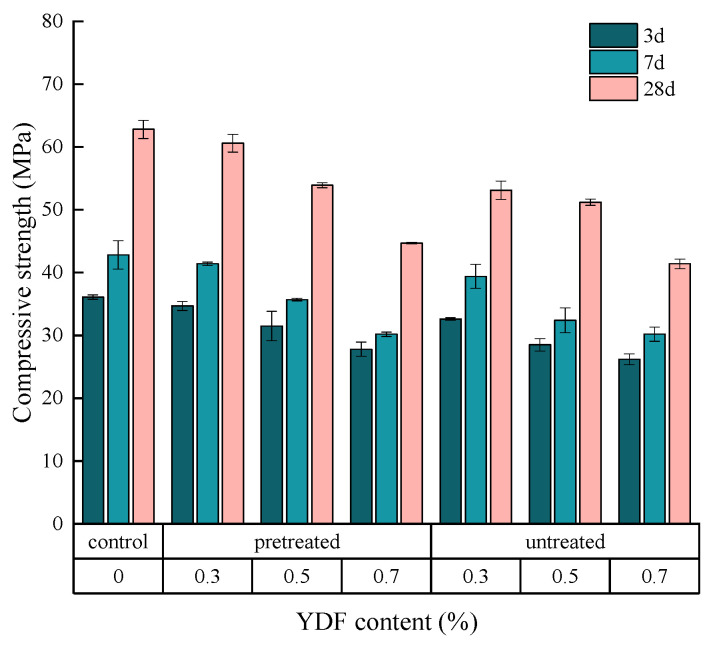
Effects of YDF content on the compressive strength of cementitious mortars.

**Figure 8 materials-16-00719-f008:**
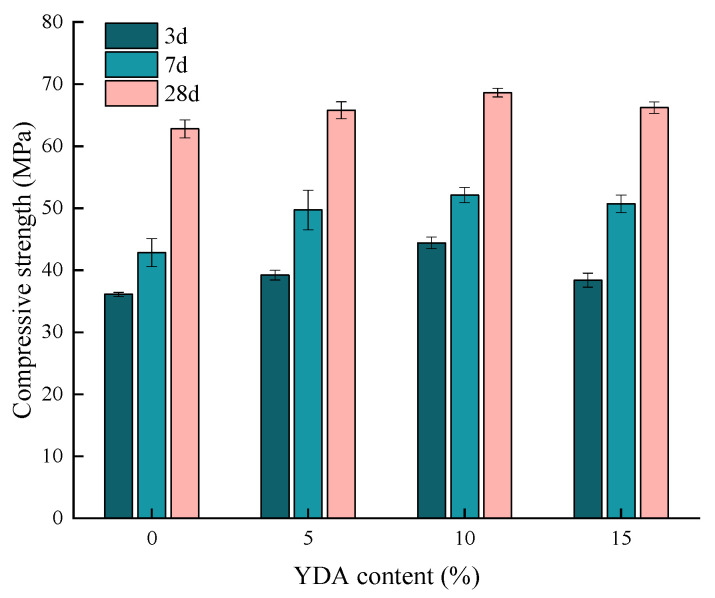
Effects of YDA content on the compressive strength of cementitious mortars.

**Figure 9 materials-16-00719-f009:**
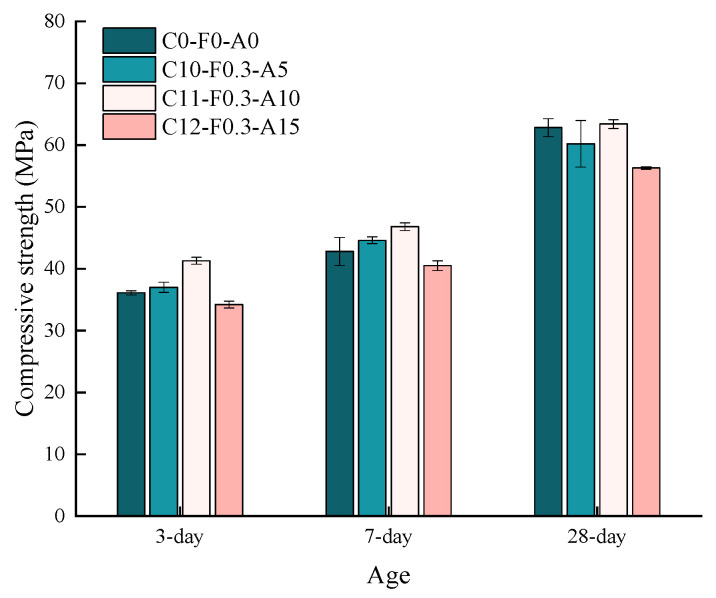
Effect of synergistic action of YDF and YDA on the compressive strength of cementitious mortars.

**Figure 10 materials-16-00719-f010:**
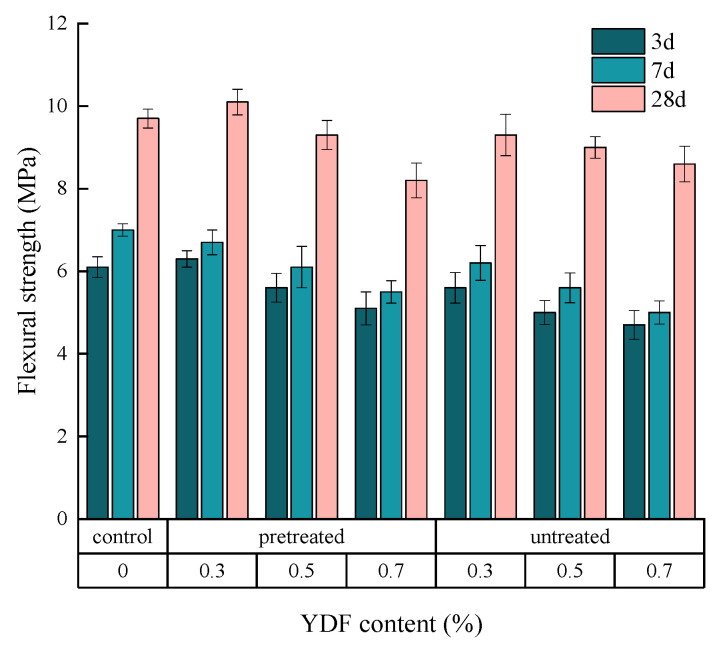
The relationship between the YDF content, the YDF type and the flexural strength of the cement mortar at different curing ages.

**Figure 11 materials-16-00719-f011:**
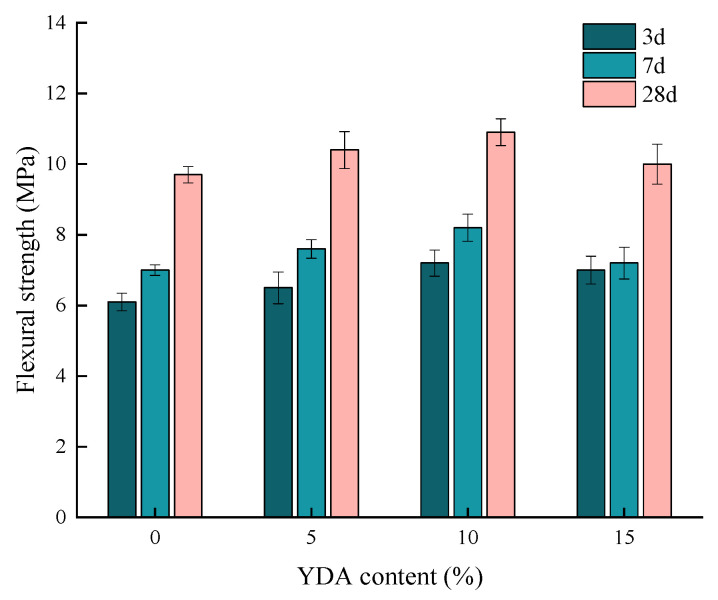
Effects of YDA content on the flexural strength of cementitious mortars.

**Figure 12 materials-16-00719-f012:**
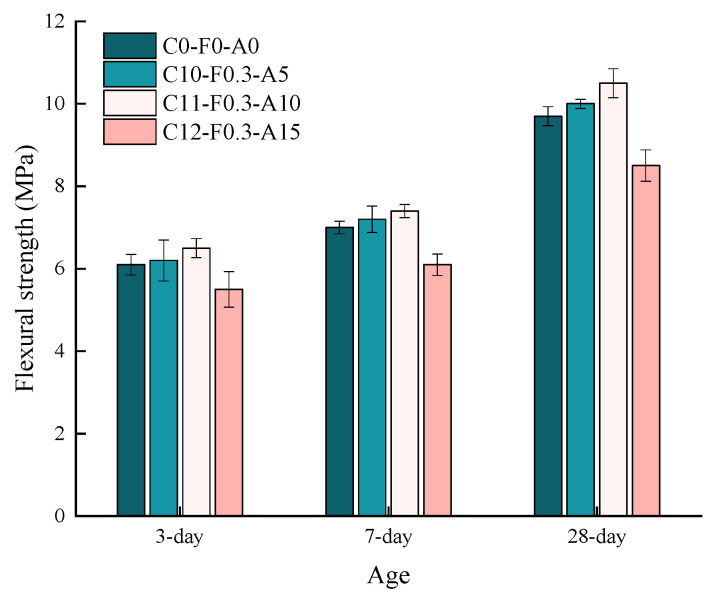
Flexural strength of cementitious mortars at different ages by the coordinated action of YDF and YDA.

**Figure 13 materials-16-00719-f013:**
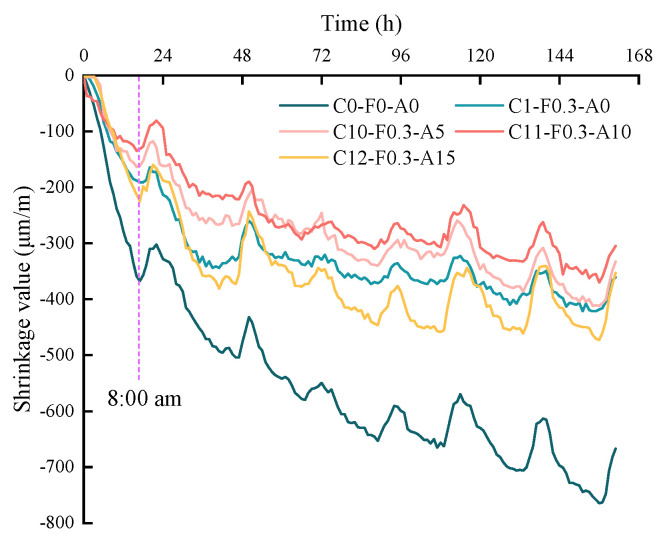
The shrinkage deformation curves of cement mortars with and without pretreated YDF.

**Figure 14 materials-16-00719-f014:**
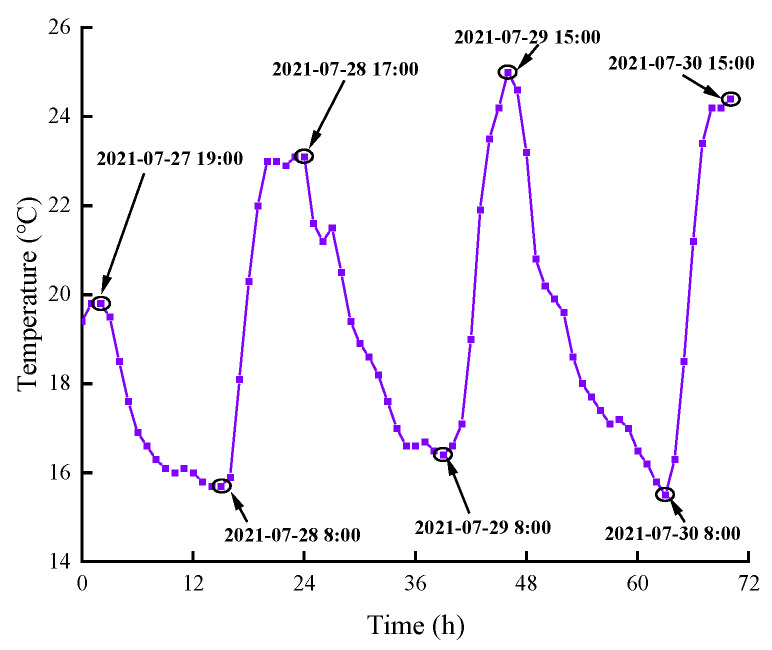
Variation curve of laboratory temperature during shrinkage experiment.

**Figure 15 materials-16-00719-f015:**
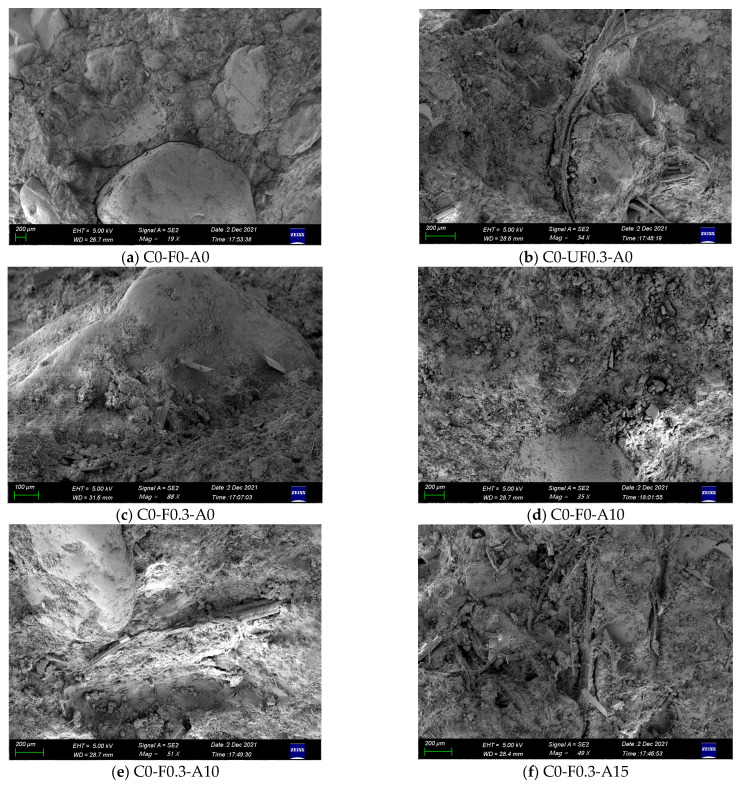
SEM image of cement mortar cured for 28 d.

**Table 1 materials-16-00719-t001:** Chemical composition of cementitious materials as detected by XRF.

Materials	Chemical Composition (wt%)								
	SiO_2_	Al_2_O_3_	CaO	Fe_2_O_3_	MgO	SO_3_	Na_2_O	Loss	Cl^−^
OPC	22.81	4.52	62.64	3.46	2.05	2.75	0.53	1.23	0.02

**Table 2 materials-16-00719-t002:** Physical properties of OPC.

Initial Setting Time (min)	Final Setting Time (min)	Density (g/cm^3^)	Specific Surface Area (m^2^/kg)	Consistency/ %	Stability (Ray Method)/mm
161	214	3.14	340	25.2	0.1

**Table 3 materials-16-00719-t003:** Physical properties of sands.

Specific Density (g/cm^3^)	Apparent Density (g/cm^3^)	Fineness Module	d_50_ (mm)	C_u_	C_c_
2.63	1.46	2.48	1.617	1.57	0.96

**Table 4 materials-16-00719-t004:** Mix proportions of YDF and YDA reinforced cementitious mortar specimen ^a^.

Type	Cement (kg/m^3^)	Sand (kg/m^3^)	Water (kg/m^3^)	YDF/%	YDA/%	SP/%	w/c
C0-F0-A0	600	1200	240	—	—	1.5	0.4
C1-F0.3-A0	600	1200	240	0.3	—	1.5	0.4
C2-F0.5-A0	600	1200	240	0.5	—	1.5	0.4
C3-F0.7-A0	600	1200	240	0.7	—	1.5	0.4
C4-UF0.3-A0	600	1200	240	0.3	—	1.5	0.4
C5-UF0.5-A0	600	1200	240	0.5	—	1.5	0.4
C6-UF0.7-A0	600	1200	240	0.7	—	1.5	0.4
C7-F0-A5	570	1200	240	—	5	1.5	0.4
C8-F0-A10	540	1200	240	—	10	1.5	0.4
C9-F0-A15	510	1200	240	—	15	1.5	0.4
C10-F0.3-A5	570	1200	240	0.3	5	1.5	0.4
C11-F0.3-A10	540	1200	240	0.3	10	1.5	0.4
C12-F0.3-A15	510	1200	240	0.3	15	1.5	0.4

^a^ F represents the cow dung fiber mixed with sodium hydroxide treatment, UF represents the cow dung fiber mixed with tap water soaking treatment.

**Table 5 materials-16-00719-t005:** Analysis of variance results for concrete properties.

	Source	Sum of Squares	DF	Mean Square	F-Value	*p*-Value
Fluidity	YDF type	10.880	1	10.880	1.272	0.311
	YDF content	603.893	3	201.298	23.530	0.002
	YDA content	2124.921	3	708.307	82.797	<0.001
28 d Compressive Strength	YDF type	26.403	1	26.403	7.327	0.042
	YDF content	337.533	3	112.511	31.221	0.001
	YDA content	31.849	3	10.616	2.946	0.138
28 d Flexural Strength	YDF type	0.019	1	0.019	0.084	0.783
	YDF content	2.438	3	0.813	3.690	0.097
	YDA content	2.261	3	0.754	3.423	0.109
156 h shrinkage strain	YDF content	99,371.868	1	99,371.868	58.494	0.005
	YDA content	44,197.950	3	14,732.650	0.224	0.864

## Data Availability

The data presented in this study are available on request from the corresponding author.

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
