# Peer review of "Synergistic Effect of Yak Dung Fiber and Yak Dung Ash on the Mechanical and Shrinkage Properties of Cement Mortar"

_materials, 2023, doi:10.3390/ma16020719_

Round 1

Reviewer 1 Report

The article presented by the authors brings an unusual study to the area of ​​civil engineering. Civil construction is one of the largest areas in which waste from other areas is incorporated.

However, the incorporation of organic material in cement matrix was the first time I saw it. The incorporation of organic material in cementitious matrices is not common, as this material undergoes chemical and physical changes over time.

The research results show that the incorporation of yak dung fiber and yak dung ash did not help in the compressive strength of the matrix, only the yak dung fiber had a small improvement in tensile strength, but it could be a result purge point, ie the normal variation of samples in an assay.

I understand that the authors did not intend to evaluate interference over time, considering, for example, years. But it is of fundamental importance that this analysis can be carried out to verify the real possibility of implantation of the material in a cement matrix.

Some points about the article need to be answered and added to the document:

1- In the introduction, little about the properties and durability of ash and manure fiber are addressed. Authors should delve deeper into these points;

2 - It is treated about the chemical characterization of the manure, but nothing about the mechanical properties, being of paramount importance for a concrete reinforcement;

3 - Table 4 presents the proportion of constituents, but it is not common to present consumption in grams. Authors must present in kg/m3;

Author Response

Dear reviewer:

Thank you for your time and valuable comments.

Best regards.

Reviewer 2 Report

In this research, the high value use of agricultural and livestock waste resources in the context of a low carbon economy is investigated. Yak dung, as a waste resource from livestock farming in the plateau, is considered to be a misplaced treasure. In this work, yak dung was processed into yak dung fiber (YDF) and yak dung ash (YDA) respectively, and the microscopic morphology of the YDF and YDA was assessed using scanning electron microscopy (SEM), the elements in the YDA were analyzed by energy dispersive X-ray spectroscopy (EDX).

Thematically the work is interesting for the researchers and professionals and the proposed manuscript is relevant to the scope of the journal.

I found it appropriate for publication in the Materials journal, but only after some modifications and clarification from the Authors.

The overall organization and structure of the manuscript are appropriate. The paper is well written and the topic is appropriate for the journal.
The aim of the paper is well described and the discussion was well approached, its results and discussion are correlated to the cited literature data.
In the introductory part, the authors give elaboration of the overall context stating the motivation and the objectives of the work, literature review of the research pathways .
The literature review is comprehensive and properly done. Perhaps a newer references could be introduced in the Introduction section in some places?

In the final paragraphs of the introductory section the authors explain what is the core of their research. However, it has to be clearly stated by the authors what is their contribution that makes the research different enough in comparison to the other authors' works and they have to further elaborate the extent of novelty in their research. The novelty of the work must be more clearly demonstrated.
The significance of the Work: Given the large number of analyzed data, this is an interesting study with a possible significant impact in this area.
Statistical interpretation of the analytical data must be more properly presented. At least an ANOVA analysis could be performed to test the influence of the variables for results presented in figures 6-13? Please check if the means were different between samples?

Other Specific Comments: The work is properly presented in terms of the language. The work presented here is very interesting and well done, it is presented in a compact manner.

The main drawback of the paper i s the extent of novelty, or the main novelty in the present work, compared to the works of other researchers? In my opinion, the authors should put additional effort to demonstrate that the present work gives a substantial contribution in the research area.

Author Response

(The authors gave the same response as above.)
